# Comparative Analysis of Volatile Compounds of Gamma-Irradiated Mutants of Rose (*Rosa hybrida*)

**DOI:** 10.3390/plants9091221

**Published:** 2020-09-17

**Authors:** Jaihyunk Ryu, Jae Il Lyu, Dong-Gun Kim, Jung-Min Kim, Yeong Deuk Jo, Si-Yong Kang, Jin-Baek Kim, Joon-Woo Ahn, Sang Hoon Kim

**Affiliations:** 1Advanced Radiation Technology Institute, Korea Atomic Energy Research Institute, Jeollabuk-do 56212, Korea; jhryu@kaeri.re.kr (J.R.); jaeil@kaeri.re.kr (J.I.L.); dgkim@kaeri.re.kr (D.-G.K.); jmkim0803@kaeri.re.kr (J.-M.K.); jyd@kaeri.re.kr (Y.D.J.); jbkim74@kaeri.re.kr (J.-B.K.); joon@kaeri.re.kr (J.-W.A.); 2Department of Horticulture, College of Industrial Sciences, Kongju National University, Yesan 32439, Korea; sykang@kongju.ac.kr

**Keywords:** rose, mutant, volatile compounds, hierarchical cluster analysis, principal component analysis

## Abstract

Roses are one of the most important floricultural crops, and their essential oils have long been used for cosmetics and aromatherapy. We investigated the volatile compound compositions of 12 flower-color mutant variants and their original cultivars. Twelve rose mutant genotypes were developed by treatment with 70 Gy of ^60^Co gamma irradiation of six commercial rose cultivars. Essential oils from the flowers of the 18 genotypes were analyzed by gas chromatography–mass spectrometry. Seventy-seven volatile compounds were detected, which were categorized into six classes: Aliphatic hydrocarbons, aliphatic alcohols, aliphatic ester, aromatic compounds, terpene alcohols, and others. Aliphatic (hydrocarbons, alcohols, and esters) compounds were abundant categories in all rose flowers. The CR-S2 mutant had the highest terpene alcohols and oil content. Three (CR-S1, CR-S3, and CR-S4) mutant genotypes showed higher ester contents than their original cultivar. Nonacosane, 2-methylhexacosane, and 2-methyltricosane were major volatile compounds among all genotypes. Hierarchical cluster analysis (HCA) of the rose genotypes gave four groups according to grouping among the 77 volatile compounds. In addition, the principal component analysis (PCA) model was successfully applied to distinguish most attractive rose lines. These findings will be useful for the selection of rose genotypes with improved volatile compounds.

## 1. Introduction

The rose belongs to the Rosaceae family, including about 200 species distributed throughout the world [1,2]. Roses are an important floricultural crop and are the most popular cut flowers because of their various floral colors and shapes [3]. Flower color is the major horticultural characteristic of rose cultivars [3,4]. A broad range of floral colors are now available in the rose after many years of cultivar development, and many studies have investigated the contributions of rose pigments [5,6,7]. Rose flower colors are caused by the presence of pigments such as anthocyanins (cyanidin, peonidin, and pelargonidin), flavonols (quercetin and kaempferol), and carotenoids (xanthophylls) [1,4,5,6,7].

Rose cultivars have developed characteristics such as long vase life, novel flower shape and color, disease tolerance, and fragrance [2,3]. Recently, many countries have made efforts to develop their own rose cultivar to produce high-quality fragrances and large amounts of oil [1,2,8]. Although wide variability in the flower has been generated through hybridization, the floriculture industry relies on a limited number of mutated traits in accordance with specific consumer expectations [3,8,9,10]. Mutation breeding involves the use of various mutagens to develop plants that exhibit a few mutated characteristics without disturbing the other characteristics of original cultivar [10,11]. Novel rose genotypes generated by radiation mutagenesis and showing improved flower traits have been developed using mutation breeding techniques [10,11,12]. In our previous work, which explored diverse variations in color and number of petals, mutations were generated by gamma rays in three spray-type (‘Lovelydia,’ ‘Yellowbabe,’ and ‘Haetsal’) and two standard-type (‘Vital’ and ‘Aqua’) rose cultivars [12].

Rose products are in high demand throughout the world. In particular, rose essential oils have wide applications in food, cosmetics, and pharmaceuticals [1,2,13]. Although there are about 18,000 rose cultivars, only a few of them are used for fragrance applications [14,15,16]. This fragrance application was mainly distributed in France, China, Bulgaria, and India [17]. Volatile compounds are important contributors to fragrance in roses and other flowers [16,17,18]. The chemical characterization of volatile compounds in flowers is of paramount importance for the identification of novel materials that have potential for industrial use [17,18]. In addition, the volatile compounds of rose are important substances in therapeutic applications because of a range of bioactivities. Rose flowers emit a strong fragrance, and various volatile compounds such as citronellol, geraniol, and nerol have been identified in rose flower extracts [2,13,15,16,18]. These volatile compounds were found to have diverse biological activities such as antioxidant, antibacterial, antitumor, antiulcer, anti-inflammatory, and anti-parkinsonism activities; they can also decrease blood cholesterol and can be used as emollients and surfactants [16,19,20,21,22].

Many methods such as hierarchical cluster analysis (HCA) and principal component analysis (PCA) methods have been applied for identification of subgroups as determined by the measure of similarity. As the HCA and PCA tools emphasize their contribution of variability, they can be important methods to speed up the breeding program [23]. The objective of the present study was to investigate and compare the volatile compounds of flower color-changing rose mutant derived from gamma rays and those of original cultivars using GC-MS, and applied to HCA and PCA to determine which cultivars would be suitable for the fragrance industry.

## 2. Results

### 2.1. Flower Morphological Characteristics

The evaluation of flower morphological characteristics is presented in Table 1 and Figure 1. Mutant cultivar CR-S7, which produces white pink petals, was developed from the ‘Aqua’ cultivar (red-purple petal). Mutant ‘CR-S2’ was derived from the black red ‘Blackbeauty’ cultivar, and showed black pink petals with white mosaic. Mutant CR-S5 had ivory petals, while its original cultivar had pale pink petals. The mutants CR-S1, CR-S3, and CR-S4, which showed orange red or red petals, were derived from the ‘Vital’ cultivar (cherry red). The CR-S6 mutant with orange petals was developed from the ‘Yellowbabe’ cultivar. Five mutant genotypes were derived from the ‘Lovelydia’ cultivar: Two showed changes in petal numbers and three showed changes in petal colors.

### 2.2. Analysis of Oil Content and Volatile Compounds

The oil content of the flower samples from the rose mutants and the original cultivars is shown in Figure 2. The results showed significant difference (*p* < 0.05) in the oil content between the different rose lines. The oil content of all rose lines ranged from 0.10 to 0.43 mL·kg^−1^, the highest values being observed for CR-S2. CR-S7, CR-S5 and CR-12 showed higher oil contents compared to those of the respective original cultivars, while CR-S1 and CR-S4 exhibited lower oil contents compared to those of the respective original cultivars.

The volatile compound constituents of the 12 rose mutants and the original cultivars were determined by gas chromatography–mass spectrometry (GC-MS). The distribution of identified volatile compounds observed from the RTX-5MS column is shown in Appendix A. The GC-MS analysis detected 77 volatile compounds in the novel rose mutant and their original cultivars, of which all compounds were tentatively identified by mass spectra and retention time, based on a NIST library similarity index greater than 90%. For individual cultivars, the number of volatile compounds ranged from 40 (CR-S10) to 50 (CR-S2 and CR-S12) with an average of 47.6.

Figure 3 shows that the volatile compounds belonged to six classes: Aliphatic hydrocarbons, aliphatic alcohols, aliphatic esters, aromatic (hydrocarbons, ester, alcohol, and ketones) compounds, acyclic terpene alcohols, and others. Aliphatic hydrocarbons were the dominant volatile compound category present in all rose flowers. The content of aliphatic hydrocarbons in the volatiles for all rose genotypes ranged from 63.1% to 80.9% with an average value of 70.1%; the highest aliphatic hydrocarbon was found in the ‘Vital’ cultivar. The highest aliphatic alcohol content (15.8%) was observed in the ‘Lovely Lydia’ cultivar and the lowest alcohol content (2.2%) occurred in the CR-S2 mutant. The aliphatic ester contents of all the genotypes ranged from 5.0% to 14.0% with the highest composition observed in the ‘Aqua.’ The content of total aromatic (hydrocarbons, alcohol, and ester) compounds in the volatiles for all rose genotypes ranged from 0.9% to 2.6% with an average value of 1.8%. The CR-S2 mutant had the highest terpene alcohol content (9.0%) while the ‘Vital’ cultivar had the lowest (0.5%). Similar comparisons were observed for mutant CR-S5 and its original ‘Haetsal’ cultivar, and for CR-S6 and its original ‘Yellowbabe’ cultivar. Mutants CR-S8, CR-S9, CR-S10, CR-S11, and CR-S12 had higher aliphatic hydrocarbon contents and lower aliphatic alcohol contents in the volatiles than the original ‘Lovely Lydia’ cultivar. The aliphatic hydrocarbon contents in the volatiles of CR-S1, CR-S3, and CR-S4 mutants were lower than those of the original ‘Vital’ cultivar. Mutants CR-S1, CR-S3, and CR-S4 showed higher aliphatic alcohol contents than the original ‘Vital’ cultivar. The aliphatic ester contents in the volatiles of CR-S1, CR-S3, and CR-S4 mutants were higher than those of the original ‘Vital’ cultivar. The terpene alcohol contents in the volatiles of CR-S2 mutants were higher than those of the original ‘Blackbeauty’ cultivar. 

Table 2 presents the ten most abundant volatile compounds detected in the GC-MS analysis of the 12 rose mutants and the original cultivars. The results revealed significant differences in the rose genotypes. Nonacosane was a dominant compound in most cultivars; its highest content (18.7%) was observed in the CR-S12 mutant and its lowest (4.4%) in the ‘Haetsal’ cultivar. The content of 2-methylhexacosane for all rose genotypes ranged from 14.0% for ‘Aqua’ to 22.8% for CR-S9, with an average content of 18.2%. The 2-methyltricosane contents for all rose genotypes ranged from 8.0% for CR-S6 and CR-S8 to 20.3% for the ‘Vital’ cultivar, with an average value of 11.4%. The tricosane contents for all genotypes ranged from 4.4% to 19.6% with the highest rate observed in the ‘Vital’ cultivar. Hentriacontane was found in the top ten volatile compounds in all cultivars except for the ‘Vital’ cultivar, with contents ranging from 2.8% for ‘Yellowbabe’ to 8.7% for CR-S8. Tetracosyl pentafluoropropionate was not observed in the top ten volatile compounds in mutant CR-S2 and the ‘Vital’ cultivar, but was found in the remaining 16 genotypes at levels between 1.8% and 6.8%. Heneicosane was observed in the top ten volatile compounds in 14 genotypes at levels between 2.0% and 7.6%, but was not observed in ‘Aqua,’ CR-S7, CR-S8, or CR-S12. Octacosanol was observed in the top ten volatile compounds in 16 rose genotypes at levels ranging from 1.8% for CR-S12 to 6.8% for the ‘Blackbeauty’ cultivar, with an average content of 3.7%. Analyses for ‘Vital’ and CR-S2 did not show octacosanol in the top ten volatile compounds. 2-Octyl-1-decanol was listed in the top ten volatile compounds in six genotypes. Octacosane was recorded in the top ten volatile compounds in nine genotypes. Hexacosane was observed in the top ten volatile compounds in the ‘Vital’ cultivar and the CR-S3 mutant. (*Z*)-14-Tricosenyl formate was detected in the top ten volatiles of the ‘Aqua’ cultivar and in the CR-S2 and CR-S12 mutants at levels of 2.7%, 4.0%, and 2.2%, respectively. The highest 1-triacontanol content was 5.8% in the CR-S2 mutant, while for the remaining cultivars, it was only listed in the top ten volatiles for ‘Aqua’ and ‘Blackbeauty’ cultivars. Other top 10 compounds included (*Z*)-9-tricosene, decane, *n*-tetracosanol, hexadecanal, tetracosane, tritriacontane, tetracosanal, octacosyl pentafluoropropionate, trifluoro-acetic acid, undecyl ester, heptacosyl heptafluorobutyrate, and dodecane.

### 2.3. Chemical Hierarchical Cluster and PCA Analysis

The results of the hierarchical cluster analysis are presented in Figure 4. The 18 rose genotypes clustered into four groups, which formed two independent supergroups (Groups I and II, and Groups III, IV, ‘CR-S1’). Group I contained four mutants (‘CR-S2,’ ‘CR-S3,’ ‘CR-S9,’ and ‘CR-S10’) and the ‘Vital’ cultivar. Group II contained two mutants (‘CR-S4’ and ‘CR-S7’) and two (‘Haetsal’ and ‘Yellowbabe’) original cultivars. Group III contained four mutants (‘CR-S5,’ ‘CR-S8,’ ‘CR-S11,’ and CR-S12) and the ‘Lovelydia’ cultivar. Group IV contained the CR-S6 mutant and two (‘Blackbeauty’ and ‘Aqua’) original cultivars. The ‘CR-S1’ mutant was found to belong to an independent group.

The chemical hierarchical cluster analysis divided the nine chemical compounds into four clusters. Cluster I contained 17 compounds, of which 5 compounds (tetracosanal, nonacosane, octacosane, (Z)-14-tricosenyl formate, and 1-triacontanol) are listed in the top ten major compounds. Cluster II contained 16 compounds, of which 5 compounds (octacosanol, tetracosyl pentafluoropropionate, tritriacontane, heneicosane and acetic acid, and trifluoroundecyl ester) are listed in the top ten major compounds. Cluster III contained 16 compounds, of which 3 compounds (octacosyl pentafluoropropionate, heneicosane, and 2-octyl-1-decanol) are listed in the top ten major compounds. Cluster IV contained 28 compounds, of which 8 compounds (dodecane, 2-methylhexacosane, decane, n-tetracosanol-1, 2-methyltricosane, (Z)-9-tricosene, hexacosane, and tricosane) are listed in the top ten major compounds.

PCA analysis was used to separate all the rose mutant lines and their original cultivars (Figure 5). Seven principal components (PC) explaining 73.1% of the total variance isolated the analyzed mutant lines and their original cultivars. PC1, which explains 16.9% of the total variance, is clearly isolated CR-S1 mutant. PC2, which explains 15.4% of the total variance, is clearly isolated CR-S10 mutant. PC3, which explains 12.6% of the total variance, is clearly isolated Haitsal cultivar. PC4, which explains 8.1% of the total variance, is not clearly isolated among the rose mutants and their original cultivars. PC5, which explains 6.9% of the total variance, is clearly isolated CR-S5. PC6, which explains 6.6% of the total variance, is clearly isolated Yellowbabe and CR-S9. PC7, which explains 6.5% of the total variance, is clearly isolated Lovelydia.

## 3. Discussion

Rose is a very popular ornamental crop, and there is always demand for new characteristics in horticulture and the cosmetics industry. These industries prosper on the back of new traits such as flower color and fragrance [1,8,22]. In this study, the 12 rose mutants, which were derived through gamma irradiation (70 Gy), had changed petal colors and numbers. Previous work has shown 70 Gy gamma irradiation of root cuttings to be an effective means of inducing mutations in rose plants [12,24]. Research on the pigmented components of rose has already been conducted in many studies. The dominant pigments in rose petals are carotenoids and anthocyanins, which provide yellow and pink colors, respectively. Typically, orange petals result as a mixture of anthocyanins and carotenoids [4,5,6,7].

The composition of volatile compounds in flower extracts is an important determinant of oil quality [18,25]. When rose breeding is undertaken for the production of oil materials, an efficient procedure must be established for regular estimation of oil composition [1,8]. Typically, the development of new cultivars in ornamental crops is conducted through hybridization. However, hybridization can result in drastic changes in volatile compound composition [8,26]. In this study, hybridization was not used to develop aroma and oil compositions in rose cultivars. Rather, mutation breeding was performed on elite cultivars so that relatively few morphological traits were altered and oil composition could be tuned [10]. Mutagenesis using radiation has been found to be effective for introducing variability in oil composition to various crops [10,27,28].

To date, about 400 different volatile compounds have been reported in rose plants, and these compounds have been categorized into major chemical groups, such as aliphatic, terpenes, aromatic, and others [29,30]. This study revealed that aliphatic hydrocarbons, aliphatic esters, and alcohols are major volatiles in all rose genotypes. Aliphatic compounds are used for industrial applications such as fragrances, paraffin, and wax [15,29]. The main rose oil paraffin contents range from 13% to 23% according to the ISO9842 rose oil standard [31]. Compared to previous reports, there were some differences in the chemical composition rates of rose oil. Oktavianawati et al. reported that they detected only aliphatic hydrocarbons (100%) in three rose cultivars [13]. However, Babu et al. reported that long-chain aliphatic hydrocarbons including nonadecane, heptadecane, 9-eicosene, and docosane accounted for about 21.23% of the volatiles, while alcohols made up as much as 68.13% of the total oil in the Himalayas rose [32]. Kazaz et al. reported that the main compound groups of rose oil are monoterpene alcohols and hydrocarbons [14]. The aliphatic hydrocarbon contents in the CR-S8 and CR-S9 mutants were about 10% higher than those in the original cultivar. By contrast, the aliphatic hydrocarbon contents in the CR-S1, CR-S3, and CR-S4 mutants were lower than those of the original cultivar. Aliphatic esters and alcohols are important compounds in food, cosmetics, and medicinal products; examples include fragrances in shampoo, perfumes, soaps and creams, aromatherapy oil, and food supplements [16,33]. In addition, aliphatic ester and alcohol compounds can also have bioactive properties, and can be used as emollients, surfactants, and antioxidants [14,16,17,19]. The aliphatic ester contents of CR-S1, CR-S3, and CR-S4 mutants were higher than those of the original cultivar. The aliphatic alcohol contents of CR-S7 mutants were higher than those of the original cultivar. The compositions of aliphatic compounds in rose essential oil are the major compounds used to evaluate the quality of the oil [33]. It is normal that the compositions of volatile compounds are influenced by factors such as genotype, climate, and harvest times [1,2,27,32,33]. However, all of the rose genotypes were grown under the same conditions, and any differences in volatile compounds were likely caused by genotype. This ability to control volatile compound content through genotype may have applications in rose breeding programs for oil materials [1,2,14]. Therefore, color changes of rose mutant genotypes may also be mediated by the accumulation of flower volatile compounds. This finding is similar to that of a previous study, which changed the volatile compound compositions in *Chrysanthemum* mutant cultivars generated by gamma irradiation [27].

The results of this study revealed that the flower extracts contained 77 compounds, of which nonacosane, 2-methylhexacosane, tricosane, hentriacontane, tetracosyl pentafluoropropionate, heneicosane, and octacosanol were the major volatile compounds among the rose genotypes. Significant differences in volatile compound composition were observed among the rose genotypes. The major volatile compounds (nonacosane, heneicosane, and tricosane) were similar to those previously reported in eight rose accessions [27]. Moreover, many studies have reported that rose oil contains long-chain aliphatic hydrocarbons, including nonadecane, heptadecane, 9-eicosene, and docosane. Nonacosane and tricosane are known to exhibit antibacterial activity [34]. 2-Methylhexacosane is reported to have antimicrobial activity and the ability to decrease blood cholesterol [35]. Hentriacontane exhibits antitumor activity and has anti-inflammatory effects through the ability to suppress NF-κB and caspase-1 activation [36]. Octacosanol is found in many plant oils and is reported to have medicinal properties. Octacosanol has important biological activities, including antioxidant, antiulcer, and anti-inflammatory activities, and is a known anti-parkinsonism agent [37]. In the CR-S3 mutant, the content of octacosanol was slightly higher than that of the original cultivar. We chose n-hexane as the extraction solvent. Previous studies reported the n-hexane extraction from rose flowers, which may contain high amounts of volatiles and showed a broad range of antioxidant [38]. This study showed that the flowers of the novel rose genotypes can be a rich source of various bioactive phytochemicals.

Hierarchical cluster analysis categorized the 12 rose mutant genotypes and six original cultivars according to their volatile compound similarities. The chemical hierarchical cluster analysis produced information that can be applied to select for chemotype in breeding programs and other useful information [27,39]. In this study, we found high levels of chemical diversity among the rose genotypes. Rose accessions could be divided into four major groups, where Group I mainly contained high levels of dodecane and lacked 2-methyltricosane, and Group II included four genotypes with high levels of nonacosane, octacosane, and tricontane and lacked 2-methyltricosane. The highest heneicosane contents were found in Group III. Group IV mainly contained high 1-pentacosanol, 1-tricosanol, and (*Z*)-9-tricosane and lacked heneicosane. These trends suggest that the aliphatic hydrocarbon and aliphatic alcohol contents of rose flowers could be used as a marker to assess chemotypes.

The PCA analysis carried out for the purposes of this study thus confirmed significant differences in the volatile compound composition of rose flower depending on the genotypes. At the same time, it indicated some common features of selected cultivars. The PCA analysis carried out for the selected genotypes, CR-S1, CR-S5, CR-S9, CR-S10, Haitsal, and Lovelydia, showed different characteristics compared to other rose genotypes. The PCA tool reduction method constitutes extracting the most important factor from the chemical analysis data and analyzing the structure of the variables [40].

Our results suggest that CR-S2, CR-S5, and CR-S1 are the most suitable sources for oil production because they contained higher oil yields or specificities of the volatile compound characteristics. These results could be applied to breeding programs to develop rose cultivars with improved volatile compounds.

## 4. Materials and Methods

### 4.1. Plant Material

The mutant genotypes of rose were generated by the treatment of root cuttings of each original cultivar with 70 Gy of gamma irradiation (^60^Co): CR-S7 mutant was derived from ‘Aqua’; CR-S2 mutant was derived from ‘Blackbeauty’; CR-S5 was derived from ‘Haetsal’; CR-S1, CR-S3, and CR-S4 mutants were derived from ‘Vital’; CR-S6 mutant was derived from ‘Yellowbabe; and CR-S8, CR-S9, CR-S10, CR-S11, and CR-S12 mutants were derived from ‘Lovely Lydia.’ These mutants were selected from flower-color variants and exhibited stable inheritance of the phenotype for V_4_ generations. The radiation mutant genotypes were grown by the Radiation Breeding Research Team at the Advanced Radiation Technology Institute, Korea Atomic Energy Research Institute. The flowers were randomly collected when fully open from the same plantation. Fresh flowers (100 g) of 18 rose genotypes and 500 mL distilled water were subjected to hydro-distillation for 4 h with n-hexane as the collecting solvent. The extracts were dried over anhydrous sodium sulfate to eliminate moisture and filtered through a polyvinylidene fluoride syringe filter (0.45 µm) for GC-MS analysis. Three replicates were used for each sample. The oil yield was estimated on a moisture-free basis.

### 4.2. GC-MS Analysis of Volatile Compounds

The volatile compound compositions were analyzed using a GC-MS (Plus-2010, Shimadzu, Kyoto, Japan) equipped with a Rtx-5MS (30 m × 0.32 mm × 50 µm, Shimadzu, Kyoto, Japan) column. The carrier gas was 99.99% high-purity helium with a column flow rate of 1.37 mL/min. Sample injection was performed in splitless mode. The oven temperature was initially set at 40 °C, and was gradually increased to 300 °C at 5 °C/min with a final hold for 5 min. The mass spectrometry parameters included: Electron-impact ionization, 70 eV; ion source temperature, 230 °C; scan range, 40–500. The identification of each compound was performed using mass spectral libraries and Kovats retention indexes (RI). The GC-MS analysis detected volatile compounds in the rose mutants and those of the original cultivars, and compounds were tentatively identified based on a NIST library similarity index greater than 90%. The retention indices of all GC peaks were calculated with retention times of C7–C40 saturated alkane standards under the same chromatographic conditions. The RI of each compound on each column was calculated using the formula; *y* and *z* are carbon numbers of alkane standards, *T*_(x)_ is the retention time of the compound, and *n* and *n* + 1 represent the retention times of the alkane standards.
(1)$$RI=100y+100\left(z-y\right)\times \left(\frac{T_{\left(x\right)}-T_{\left(n\right)}}{T_{n+1}-T_{n}}\right)$$

### 4.3. Statistical Analysis

The chemical analysis data were subjected to analysis of variance using a multiple comparisons method with the SPSS (SPSS Inc., Chicago, IL, USA) version 12 statistical software package. Differences were considered significant at the 5% level. When the treatment effect was significant, means were separated using Duncan’s Multiple Range Test.

Clustering analysis of samples from the flowers of the 18 rose genotypes was performed using the complete linkage method in the SPSS software (ver. 25, SPSS Inc., Chicago, IL, USA). The volatile compounds were visualized as *z*-values in the heatmap.

PCA was used to detect clustering and to investigate possible relationships between volatile compounds. The data variability of the PC of the total volatile compounds was extracted by the accumulated variance levels at 70% based on the PC of eigenvalues higher than 1 using SPSS (ver. 25, SPSS Inc., Chicago, IL, USA).

## 5. Conclusions

GC-MS analysis of 12 mutated rose genotypes obtained through gamma-irradiation and their six original cultivars identified 77 volatile compounds that were grouped into five functional compound categories: Aliphatic hydrocarbons, aliphatic alcohols, aliphatic ester, aromatic compounds, terpene alcohols, and others. Three mutant genotypes derived from the ‘Vital’ cultivar showed increased ester content and decreased hydrocarbon content. Chemical hierarchical cluster analysis revealed that the hydrocarbon and alcohol content of rose flowers could be used as key markers to assess chemotypes. In PCA analysis, these were possible to select an elite line that could not be confirmed in cluster analysis. Thus, we construe that these volatile compounds are useful for classification and identification of rose mutant genotypes. In addition, our research suggests that the generation of novel rose genotypes by radiation breeding to give enhanced content of various bioactive phytochemicals may be an effective route to resources for use in the food and cosmetics industries, horticulture, and aromatherapy.

## Figures and Tables

**Figure 1 plants-09-01221-f001:**
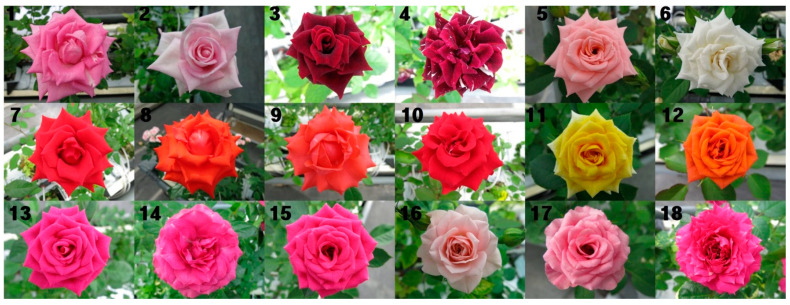
Profile of rose mutant genotypes used in this study. **1**: Aqua, **2**: CR-S7 (Aqua mutant), **3**: Blackbeauty, **4**: CR-S2 (Blackbeauty mutant), **5**: Haetsal, **6**: CR-S5 (Haetsal mutant), **7**: Vital, **8**: CR-S1 (Vital mutant), **9**: CR-S3 (Vital mutant), **10**: CR-S4 (Vital mutant), **11**: Yellowbabe, **12**: CR-S6 (Yellowbabe mutant), **13**: Lovelydia, **14**: CR-S8 (Lovelydia mutant), **15**: CR-S9 (Lovelydia mutant), **16**: CR-S10 (Lovelydia mutant), **17**: CR-S11 (Lovelydia mutant), **18**: CR-S12 (Lovelydia mutant).

**Figure 2 plants-09-01221-f002:**
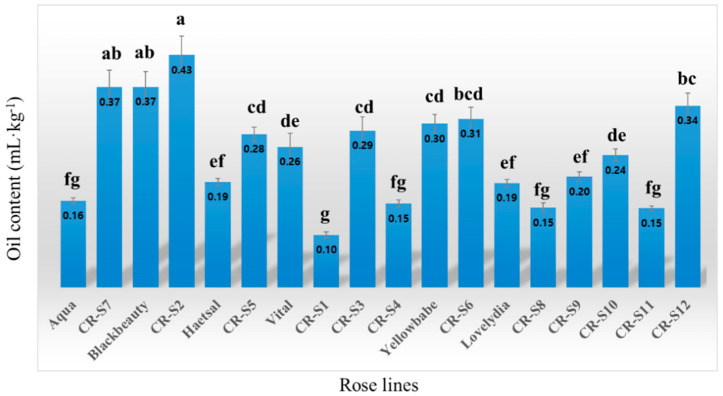
Oil yield of flower for rose mutants and their original cultivars. The letters above each point indicate a significant difference at the 5% level (Duncan’s multiple range tests, *n* = 3).

**Figure 3 plants-09-01221-f003:**
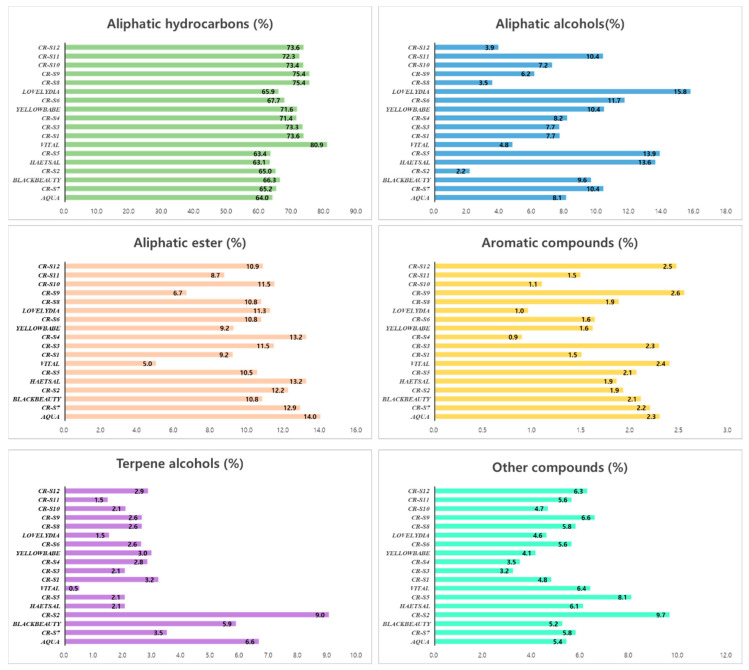
Relative composition of volatile compounds released from the flower of rose mutant and those of original cultivars.

**Figure 4 plants-09-01221-f004:**
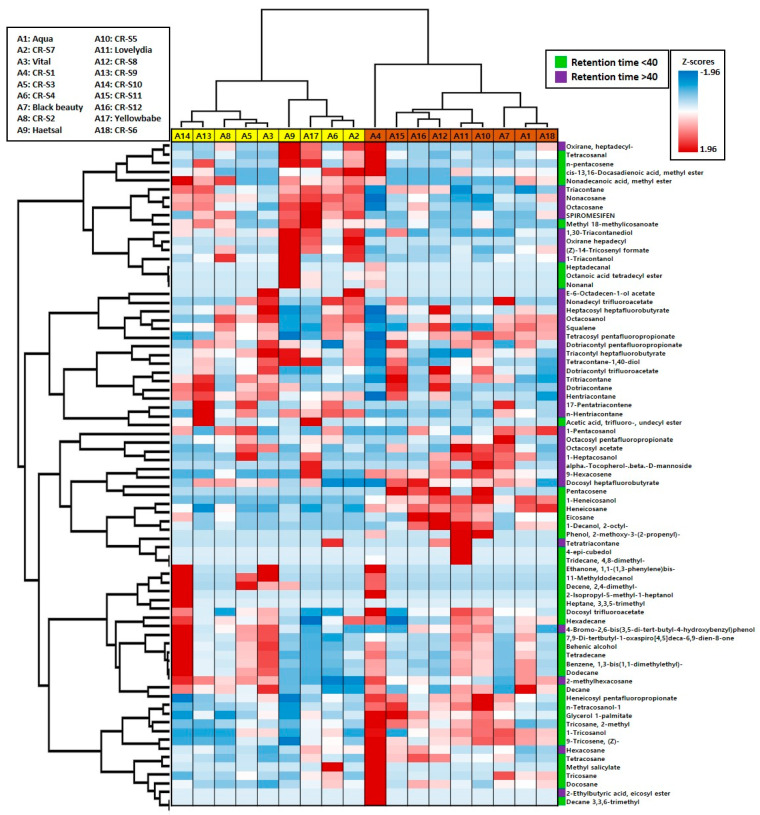
Hierarchical cluster analysis of 12 rose mutant and their original cultivars.

**Figure 5 plants-09-01221-f005:**
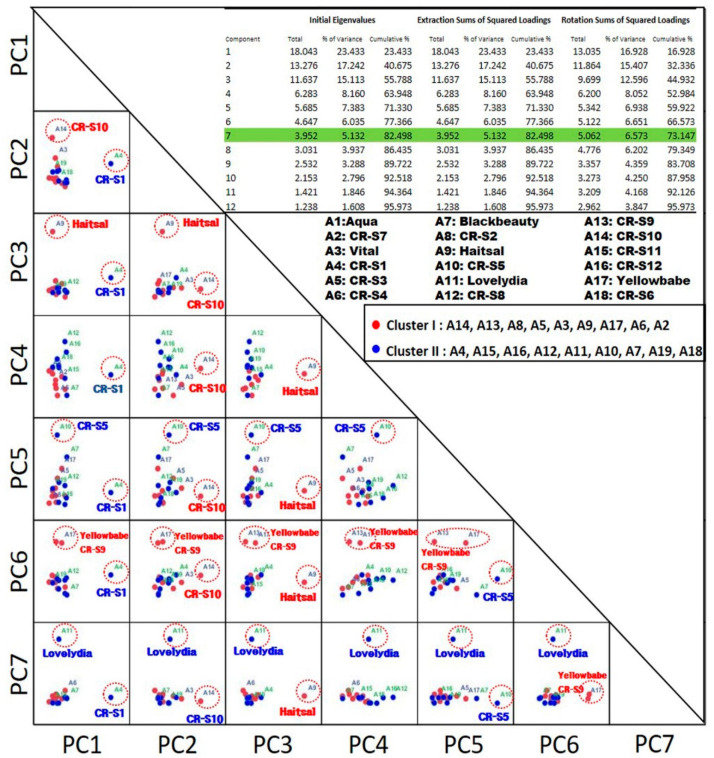
Principal component analysis (PCA) score plot of volatile compounds of 12 rose mutants and their original cultivars.

**Table 1 plants-09-01221-t001:** Origin and morphological characteristics of Rose genotypes used in this study.

No.	Names	Flower Color	Petal Numbers	Original Cultivars
1	Aqua	Red-purple (N62A *)	Normal **	
2	CR-S7	White pink (N62D)	Low	Aqua
3	Blackbeauty	Black red (N46B)	Normal	
4	CR-S2	Black pink (N46B) with white mosaic	Normal	Blackbeauty
5	Haetsal	Pale pink (N49C)	Normal	
6	CR-S5	Ivory (N11D)	Normal	Haetsal
7	Vital	Cherry red (N41B)	Normal	
8	CR-S1	Orang red (N30C)	Normal	Vital
9	CR-S3	Orang red (N32B)	Normal	Vital
10	CR-S4	Red (N41A)	Normal	Vital
11	Yellowbabe	Yellow (N9A)	Normal	
12	CR-S6	Orange (N25B)	Normal	Yellowbabe
13	Lovelydia	Red purple (N74B)	Normal	
14	CR-S8	Red purple (N74D)	Normal	Lovelydia
15	CR-S9	Red purple (N74B)	High	Lovelydia
16	CR-S10	Light beige (N36C)	Normal	Lovelydia
17	CR-S11	Orange purple (N68C)	Normal	Lovelydia
18	CR-S12	Red purple with white mosaic (N74C)	High	Lovelydia

***** The royal horticultural society’s color chart numbers, ****** Low: Under twenty, Normal: 21–40, High: Over forty.

**Table 2 plants-09-01221-t002:** Top 10 volatile compounds identified from the rose mutant and those of original cultivars.

Line Names	No.	RT	Compound Names	MF	RI	MW	Rate (%)
Aqua	1	42.01	2-methylhexacosane	C_27_H_56_	2700	380	14.0
	2	44.90	Nonacosane	C_29_H_60_	2900	408	13.5
	3	38.81	2-methyltricosane	C_24_H_50_	2541	338	10.2
	4	47.65	Hentriacontane	C_31_H_64_	3100	436	7.7
	5	45.27	Octacosanol	C_28_H_58_O	2927	410	6.5
	6	35.39	Tricosane	C_23_H_48_	2300	324	6.1
	7	42.33	Tetracosyl pentafluoropropionate	C_27_H_49_F_5_O_2_	2710	500	4.7
	8	45.10	1-Triacontanol	C_30_H_42_O	2915	438	3.6
	9	43.43	Octacosane	C_28_H_58_	2800	394	3.1
	10	42.96	(Z)-14-Tricosenyl formate	C_24_H_46_O_2_	2731	366	2.7
CR-S7	1	42.01	2-methylhexacosane	C_27_H_56_	2700	380	19.6
	2	44.90	Nonacosane	C_29_H_60_	2900	408	10.4
	3	38.81	2-methyltricosane	C_24_H_50_	2541	338	9.9
	4	47.65	Hentriacontane	C_31_H_64_	3100	436	8.0
	5	45.27	Octacosanol	C_28_H_58_O	2927	410	6.4
	6	35.39	Tricosane	C_23_H_48_	2300	324	6.1
	7	42.33	Tetracosyl pentafluoropropionate	C_27_H_49_F_5_O_2_	2710	500	5.0
	8	48.01	Heptacosyl heptafluorobutyrate	C_31_H_55_F_7_O_2_	3128	592	2.0
	9	50.22	Tritriacontane	C_33_H_68_	3300	464	1.9
	10	10.17	Dodecane	C_12_H_26_	1200	170	1.9
Blackbeauty	1	42.01	2-methylhexacosane	C_27_H_56_	2700	380	18.7
	2	44.90	Nonacosane	C_29_H_60_	2900	408	14.6
	3	38.81	2-methyltricosane	C_24_H_50_	2541	338	9.3
	4	45.27	Octacosanol	C_28_H_58_O	2927	410	6.8
	5	47.65	Hentriacontane	C_31_H_64_	3100	436	5.9
	6	35.39	Tricosane	C_23_H_48_	2300	324	5.8
	7	42.33	Tetracosyl pentafluoropropionate	C_27_H_49_F_5_O_2_	2710	500	4.9
	8	31.68	Heneicosane	C_21_H_42_	2100	296	4.4
	9	45.10	1-Triacontanol	C_30_H_42_O	2915	438	3.6
	10	43.43	Octacosane	C_28_H_58_	2800	394	1.8
CR-S2	1	42.01	2-methylhexacosane	C_27_H_56_	2700	380	17.6
	2	44.90	Nonacosane	C_29_H_60_	2900	408	16.9
	3	38.81	2-methyltricosane	C_24_H_50_	2541	338	8.2
	4	45.10	1-Triacontanol	C_30_H_42_O	2915	438	5.8
	5	47.65	Hentriacontane	C_31_H_64_	3100	436	5.3
	6	35.39	Tricosane	C_23_H_48_	2300	324	4.8
	7	36.35	Octanoic acid tetradecyl ester	C_22_H_44_O_2_	2356	340	4.6
	8	42.96	(Z)-14-Tricosenyl formate	C_24_H_46_O_2_	2731	366	4.0
	9	43.43	Octacosane	C_28_H_58_	2800	394	3.4
	10	31.68	Heneicosane	C_21_H_42_	2100	296	3.3
Haetsal	1	42.01	2-methylhexacosane	C_27_H_56_	2700	380	16.4
	2	38.81	2-methyltricosane	C_24_H_50_	2541	338	14.8
	3	42.33	Tetracosyl pentafluoropropionate	C_27_H_49_F_5_O_2_	2710	500	6.8
	4	35.39	Tricosane	C_23_H_48_	2300	324	6.6
	5	31.68	Heneicosane	C_21_H_42_	2100	296	5.7
	6	27.62	1-Decanol, 2-octyl-	C_18_H_38_O	1903	270	5.0
	7	47.65	Hentriacontane	C_31_H_64_	3100	436	4.9
	8	44.90	Nonacosane	C_29_H_60_	2900	408	4.4
	9	39.12	n-Tetracosanol-1	C_24_H_50_O	2551	354	3.9
	10	45.27	Octacosanol	C_28_H_58_O	2927	410	2.3
CR-S5	1	42.01	2-methylhexacosane	C_27_H_56_	2700	380	17.5
	2	38.81	2-methyltricosane	C_24_H_50_	2541	338	13.0
	3	27.62	1-Decanol, 2-octyl-	C_18_H_38_O	1903	270	6.7
	4	31.68	Heneicosane	C_21_H_42_	2100	296	6.6
	5	35.39	Tricosane	C_23_H_48_	2300	324	6.4
	6	44.90	Nonacosane	C_29_H_60_	2900	408	5.6
	7	42.33	Tetracosyl pentafluoropropionate	C_27_H_49_F_5_O_2_	2710	500	4.9
	8	47.65	Hentriacontane	C_31_H_64_	3100	436	3.8
	9	39.12	n-Tetracosanol-1	C_24_H_50_O	2551	354	2.5
	10	45.27	Octacosanol	C_28_H_58_O	2927	410	2.4
Vital	1	42.01	2-methylhexacosane	C_27_H_56_	2700	380	22.0
	2	38.81	2-methyltricosane	C_24_H_50_	2541	338	20.3
	3	35.39	Tricosane	C_23_H_48_	2300	324	19.6
	4	31.68	Heneicosane	C_21_H_42_	2100	296	5.6
	5	40.38	Hexacosane	C_26_H_54_	2600	366	3.4
	6	35.71	(Z)-9-Tricosene,	C_23_H_46_	2319	322	3.1
	7	39.76	Tetracosanal	C_24_H_48_O	2570	352	2.8
	8	39.12	n-Tetracosanol-1	C_24_H_50_O	2551	354	2.6
	9	37.10	Tetracosane	C_24_H_50_	2400	338	2.4
	10	15.66	Tetradecane	C_14_H_30_	1400	198	1.5
CR-S1	1	42.01	2-methylhexacosane	C_27_H_56_	2700	380	20.3
	2	44.90	Nonacosane	C_29_H_60_	2900	408	13.7
	3	38.81	2-methyltricosane	C_24_H_50_	2541	338	10.9
	4	35.39	Tricosane	C_23_H_48_	2300	324	8.0
	5	47.65	Hentriacontane	C_31_H_64_	3100	436	7.0
	6	42.33	Tetracosyl pentafluoropropionate	C_27_H_49_F_5_O_2_	2710	500	4.8
	7	45.27	Octacosanol	C_28_H_58_O	2927	410	4.1
	8	50.22	Tritriacontane	C_33_H_68_	3300	464	2.3
	9	31.68	Heneicosane	C_21_H_42_	2100	296	2.0
	10	43.43	Octacosane	C_28_H_58_	2800	394	1.7
CR-S3	1	44.90	Nonacosane	C_29_H_60_	2900	408	16.5
	2	42.01	2-methylhexacosane	C_27_H_56_	2700	380	14.6
	3	38.81	2-methyltricosane	C_24_H_50_	2541	338	12.0
	4	35.39	Tricosane	C_23_H_48_	2300	324	9.6
	5	47.65	Hentriacontane	C_31_H_64_	3100	436	6.3
	6	45.27	Octacosanol	C_28_H_58_O	2927	410	5.7
	7	42.33	Tetracosyl pentafluoropropionate	C_27_H_49_F_5_O_2_	2710	500	5.5
	8	43.43	Octacosane	C_28_H_58_	2800	394	2.9
	9	31.68	Heneicosane	C_21_H_42_	2100	296	2.8
	10	40.38	Hexacosane	C_26_H_54_	2600	366	1.5
CR-S4	1	42.01	2-methylhexacosane	C_27_H_56_	2700	380	14.5
	2	38.81	2-methyltricosane	C_24_H_50_	2541	338	13.5
	3	35.39	Tricosane	C_23_H_48_	2300	324	12.6
	4	44.90	Nonacosane	C_29_H_60_	2900	408	11.0
	5	42.33	Tetracosyl pentafluoropropionate	C_27_H_49_F_5_O_2_	2710	500	5.9
	6	47.65	Hentriacontane	C_31_H_64_	3100	436	4.8
	7	45.27	Octacosanol	C_28_H_58_O	2927	410	4.1
	8	31.68	Heneicosane	C_21_H_42_	2100	296	3.7
	9	35.71	(Z)-9-Tricosene,	C_23_H_46_	2319	322	2.3
	10	49.54	Octacosyl pentafluoropropionate	C_31_H_57_F_5_O_2_	3247	556	2.2
Yellowbabe	1	42.01	2-methylhexacosane	C_27_H_56_	2700	380	19.8
	2	44.90	Nonacosane	C_29_H_60_	2900	408	14.8
	3	35.39	Tricosane	C_23_H_48_	2300	324	9.6
	4	38.81	2-methyltricosane	C_24_H_50_	2541	338	9.0
	5	31.68	Heneicosane	C_21_H_42_	2100	296	7.6
	6	45.27	Octacosanol	C_28_H_58_O	2927	410	5.4
	7	42.33	Tetracosyl pentafluoropropionate	C_27_H_49_F_5_O_2_	2710	500	5.3
	8	47.65	Hentriacontane	C_31_H_64_	3100	436	2.8
	9	27.62	1-Decanol, 2-octyl-	C_18_H_38_O	1903	270	2.7
	10	35.71	(Z)-9-Tricosene,	C_23_H_46_	2319	322	1.7
CR-S6	1	42.01	2-methylhexacosane	C_27_H_56_	2700	380	20.3
	2	44.90	Nonacosane	C_29_H_60_	2900	408	9.9
	3	35.39	Tricosane	C_23_H_48_	2300	324	8.7
	4	38.81	2-methyltricosane	C_24_H_50_	2541	338	8.0
	5	31.68	Heneicosane	C_21_H_42_	2100	296	6.9
	6	45.27	Octacosanol	C_28_H_58_O	2927	410	5.1
	7	42.33	Tetracosyl pentafluoropropionate	C_27_H_49_F_5_O_2_	2710	500	4.8
	8	47.65	Hentriacontane	C_31_H_64_	3100	436	4.7
	9	27.62	1-Decanol, 2-octyl-	C_18_H_38_O	1903	270	3.0
	10	35.71	(Z)-9-Tricosene	C_23_H_46_	2319	322	2.1
Lovelydia	1	42.01	2-methylhexacosane	C_27_H_56_	2700	380	16.8
	2	38.81	2-methyltricosane	C_24_H_50_	2541	338	10.7
	3	27.62	1-Decanol, 2-octyl-	C_18_H_38_O	1903	270	7.9
	4	44.90	Nonacosane	C_29_H_60_	2900	408	7.8
	5	35.39	Tricosane	C_23_H_48_	2300	324	6.7
	6	47.65	Hentriacontane	C_31_H_64_	3100	436	6.6
	7	31.68	Heneicosane	C_21_H_42_	2100	296	6.0
	8	42.33	Tetracosyl pentafluoropropionate	C_27_H_49_F_5_O_2_	2710	500	4.9
	9	45.27	Octacosanol	C_28_H_58_O	2927	410	4.2
	10	50.22	Tritriacontane	C_33_H_68_	3300	464	2.4
CR-S8	1	42.01	2-methylhexacosane	C_27_H_56_	2700	380	20.7
	2	44.90	Nonacosane	C_29_H_60_	2900	408	17.5
	3	47.65	Hentriacontane	C_31_H_64_	3100	436	8.7
	4	38.81	2-methyltricosane	C_24_H_50_	2541	338	8.0
	5	35.39	Tricosane	C_23_H_48_	2300	324	6.5
	6	43.43	Octacosane	C_28_H_58_	2800	394	3.0
	7	50.22	Tritriacontane	C_33_H_68_	3300	464	2.9
	8	42.33	Tetracosyl pentafluoropropionate	C_27_H_49_F_5_O_2_	2710	500	2.3
	9	13.54	Acetic acid, trifluoroundecyl ester	C_13_H_23_F_3_O_2_	1343	268	2.2
	10	45.27	Octacosanol	C_28_H_58_O	2927	410	2.0
CR-S9	1	42.01	2-methylhexacosane	C_27_H_56_	2700	380	22.8
	2	44.90	Nonacosane	C_29_H_60_	2900	408	15.9
	3	38.81	2-methyltricosane	C_24_H_50_	2541	338	8.4
	4	47.65	Hentriacontane	C_31_H_64_	3100	436	7.6
	5	35.39	Tricosane	C_23_H_48_	2300	324	4.4
	6	31.68	Heneicosane	C_21_H_42_	2100	296	4.1
	7	43.43	Octacosane	C_28_H_58_	2800	394	2.7
	8	45.27	Octacosanol	C_28_H_58_O	2927	410	2.1
	9	10.17	Dodecane	C_12_H_26_	1200	170	2.0
	10	42.33	Tetracosyl pentafluoropropionate	C_27_H_49_F_5_O_2_	2710	500	1.8
CR-S10	1	42.01	2-methylhexacosane	C_27_H_56_	2700	380	17.9
	2	38.81	2-methyltricosane	C_24_H_50_	2541	338	14.7
	3	44.90	Nonacosane	C_29_H_60_	2900	408	9.2
	4	47.65	Hentriacontane	C_31_H_64_	3100	436	8.4
	5	35.39	Tricosane	C_23_H_48_	2300	324	6.6
	6	42.33	Tetracosyl pentafluoropropionate	C_27_H_49_F_5_O_2_	2710	500	4.7
	7	31.68	Heneicosane	C_21_H_42_	2100	296	4.6
	8	50.22	Tritriacontane	C_33_H_68_	3300	464	3.3
	9	39.12	n-Tetracosanol-1	C_24_H_50_O	2551	354	3.0
	10	45.27	Octacosanol	C_28_H_58_O	2927	410	2.0
CR-S11	1	42.01	2-methylhexacosane	C_27_H_56_	2700	380	18.3
	2	38.81	2-methyltricosane	C_24_H_50_	2541	338	12.4
	3	44.90	Nonacosane	C_29_H_60_	2900	408	12.4
	4	35.39	Tricosane	C_23_H_48_	2300	324	7.0
	5	31.68	Heneicosane	C_21_H_42_	2100	296	6.7
	6	47.65	Hentriacontane	C_31_H_64_	3100	436	5.0
	7	27.62	1-Decanol, 2-octyl-	C_18_H_38_O	1903	270	4.9
	8	42.33	Tetracosyl pentafluoropropionate	C_27_H_49_F_5_O_2_	2710	500	3.6
	9	45.27	Octacosanol	C_28_H_58_O	2927	410	3.0
	10	43.43	Octacosane	C_28_H_58_	2800	394	2.4
CR-S12	1	44.90	Nonacosane	C_29_H_60_	2900	408	18.7
	2	42.01	2-methylhexacosane	C_27_H_56_	2700	380	16.5
	3	38.81	2-methyltricosane	C_24_H_50_	2541	338	10.9
	4	35.39	Tricosane	C_23_H_48_	2300	324	8.4
	5	47.65	Hentriacontane	C_31_H_64_	3100	436	5.7
	6	43.43	Octacosane	C_28_H_58_	2800	394	3.9
	7	42.96	(Z)-14-Tricosenyl formate	C_24_H_46_O_2_	2731	366	2.2
	8	42.33	Tetracosyl pentafluoropropionate	C_27_H_49_F_5_O_2_	2710	500	2.0
	9	39.76	Tetracosanal	C_24_H_48_O	2570	352	2.0
	10	45.27	Octacosanol	C_28_H_58_O	2927	410	1.8

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
