# Peer review of "Comparative Analysis of Volatile Compounds of Gamma-Irradiated Mutants of Rose (Rosa hybrida)"

_plants, 2020, doi:10.3390/plants9091221_

Round 1

Reviewer 1 Report

The manuscript deals with interesting topic and the results may be useful in the development of new roses varieties.The MS is well written and presented. However two comments may improve the presentation and the conclusions.

Figure 1 illustrating the relative proportions of the major volatile compounds is to small and may be enlarged to be visible at the normal magnification of the page. The parts of each figure may be given different colors to make comparison more obvious.

The authors concluded that the results are useful in the selection of rose genotypes with improved volatile compounds and valued the use of gamma radiations in producing stable mutants. However, more specific recommendations of certain varieties for producing specific traits will add more values to he results of this useful article 

Author Response

Dear editor and reviewers
On behalf of my co-authors, we thank you for giving us an opportunity to revise our manuscript, and we appreciate the editor and reviewers for your positive and constructive comments and suggestions on our manuscript entitled; Comparative Analysis of Volatile Compounds of Gamma-Irradiated Mutants of Rose (Rosa hybrida)

We have reviewed our manuscript carefully, and tried our best to improve our manuscript and made some changes in the manuscript based on the editor and reviewers’ comments. We marked the changes in red in our revised paper. We appreciate for editor and reviewers’ warm work, and hope that the correction will meet with approval. The main corrections in our manuscript and the responds to the reviewer’s comments are as follows:

The manuscript deals with interesting topic and the results may be useful in the development of new roses varieties. The MS is well written and presented. However, two comments may improve the presentation and the conclusions.
- Thank you

Figure 1 illustrating the relative proportions of the major volatile compounds is to small and may be enlarged to be visible at the normal magnification of the page. The parts of each figure may be given different colors to make comparison more obvious.
- We changed Fig. 2 and given different colors

2. The authors concluded that the results are useful in the selection of rose genotypes with improved volatile compounds and valued the use of gamma radiations in producing stable mutants. However, more specific recommendations of certain varieties for producing specific traits will add more values to the results of this useful article
- We added producing stable mutants on discussion

Reviewer 2 Report

in: Abstract

- line 18: the sentence “Seventy-seven volatile…” is incorrect, because the fragrances and flavor materials are arranged according to the Beilstein system; aliphatic (hydrocarbons and oxygen-containing compounds), terpenes (hydrocarbons and oxygen-containing compounds), aromatic (hydrocarbons and oxygen-containing compounds), and others (see Bauer et al., 2001).

- line 19: “Hydrocarbons, alcohols and esters were major component in all rose essential oil” should be written instead of “Hydrocarbons, alcohols and esters were major component in all rose flowers”.

- it should be described what “major compounds” are.

- some corrections of the text are necessary, because it is not clear enough now.

in: Keywords

- some corrections of the text are necessary, because it is not clear enough now.

in: Introduction

- I think that more information about the rose, oils and extracts (importance of the plant for the region, chemical composition and their application) is necessary to be referred to.

- the aim of the paper is not clear.

in: Results

- line 88: Table 1 should be written instead of Table S1.

- the information about the yield, color, and odor of the essential oil for each sample is missing.

- in Figure 1: the classification of the compounds is incorrect.

- in Table 2: it should be given the chemical composition of the essential oils, RT, and RI. Data about terpenes is missing.

- in the text and in Table 2 the information about the total identified compounds is missing.

in: Discussion

- the explanation about different chemical composition of the samples is missing.

- all data about chemical composition of the rose essential oils should be compared to the data from other authors.

- line 199: flower essential oils should be written instead of “flower extracts”.

- line 213: [8 or 9 or 25] should be written instead of [Rusanov ??].

- the classification of the compounds is incorrect.

in Conclusions

- some corrections of the text are necessary, because it is not clear enough now.

in: Material and Methods

- the origin of the rose should be described.

- the moisture content (%) of the plant, the method for its determination and the conditions of the drying should be specified. The essential oil yield should be calculated in % to absolute dry weight.

- the information about the condition of hydro distillation of the rose flowers is missing.

- it should be explained why the solvent n-hexane is used, and n-hexane should be written instead of hexane.

- line 263: the equation should be excluded.

in: References

- the list of references should be written according to the guideline for authors.

Author Response

ear editor and reviewers
On behalf of my co-authors, we thank you for giving us an opportunity to revise our manuscript, and we appreciate the editor and reviewers for your positive and constructive comments and suggestions on our manuscript entitled; Comparative Analysis of Volatile Compounds of Gamma-Irradiated Mutants of Rose (Rosa hybrida)

We have reviewed our manuscript carefully, and tried our best to improve our manuscript and made some changes in the manuscript based on the editor and reviewers’ comments. We marked the changes in red in our revised paper. We appreciate for editor and reviewers’ warm work, and hope that the correction will meet with approval. The main corrections in our manuscript and the responds to the reviewer’s comments are as follows:

Reviewer 3 Report

In the manuscript submitted to PLANTS (891701) authors compare the volatile compounds of two kinds of Rose. This reviewer suggest the publication in PLANTS after major revision.

In its present form, manuscrit must be considered a short communication.

Theme is interesting, techniques are the adequated to solve this kind of analysis, but work fails in the validation and application parts.

The use of hiarechil tools is very porweful in this kind of studies, but autors must compare its results by using another methods as principal component analysis (PCA).

Application is very poor.

Minor comments:

* Please correct some typos.

* In Experimental, for Instrumentation, Materials and Reagents, or Programs and Databases (as SPSS, Excel, and others) ever, Product (Manufacturer, City, Country), in this order and format. Please correct in some places. In the case of USA products: Product (Manufacturer, City, State, USA).

* The number of Figures and Talbes is adequated.

* The Novelty Statement is well formulated./is unclear.

* The Highlights describes adequatly the manuscript./don't describes adequatly the manuscript.

* The Graphical Abstract describes adequatly the manuscript./don't describes adequatly the manuscript

Author Response

Reviewer #3
In the manuscript submitted to PLANTS (891701) authors compare the volatile compounds of two kinds of Rose. This reviewer suggests the publication in PLANTS after major revision.

In its present form, manuscrit must be considered a short communication.

 We rewritten in the manuscript based on the reviewers’ comments

Theme is interesting, techniques are the adequated to solve this kind of analysis, but work fails in the validation and application parts.

 We rewritten in the manuscript based on the reviewers’ comments

The use of hiarechil tools is very porweful in this kind of studies, but autors must compare its results by using another methods as principal component analysis (PCA).

- We added it

Application is very poor.

Minor comments:

* Please correct some typos.
- We changed it

* In Experimental, for Instrumentation, Materials and Reagents, or Programs and Databases (as SPSS, Excel, and others) ever, Product (Manufacturer, City, Country), in this order and format. Please correct in some places. In the case of USA products: Product (Manufacturer, City, State, USA).
-- We changed it
* The number of Figures and Talbes is adequated.
-- We changed it
* The Novelty Statement is well formulated./is unclear.
- We changed it
* The Highlights describes adequatly the manuscript./don't describes adequatly the manuscript.
- We changed it
* The Graphical Abstract describes adequatly the manuscript./don't describes adequatly the manuscript 
- We changed it

Reviewer 4 Report

The paper is interesting, but some issues should be explained:

Line 66 – is: applied for identifies – should be: applied for identification?

Line 265 – Group V ? The authors write that HCA allows for the division of the studied varieties into 4 groups. So where does group V come from?

How was the PCA analysis optimized? Did the authors generate scree plots in PCA?

Usually, PC1 contains the largest share of variability, and the subsequent components are smaller and smaller. So what are the PC1 vs. PC3, PC1 vs PC4 etc? If they are included in the work, refer to them in the discussion.

Figure 5 is unreadable – so small graphs

Discussion (lines  - 182 -190) should be expanded.

Author Response

Dear reviewers #4

On behalf of my co-authors, we thank you for giving us an opportunity to revise our manuscript, and we appreciate the editor and reviewers for your positive and constructive comments and suggestions on our manuscript entitled; Comparative Analysis of Volatile Compounds of Gamma-Irradiated Mutants of Rose (Rosa hybrida)

We have reviewed our manuscript carefully, and tried our best to improve our manuscript and made some changes in the manuscript based on the editor and reviewers’ comments. We marked the changes in red in our revised paper. We appreciate for editor and reviewers’ warm work, and hope that the correction will meet with approval. The main corrections in our manuscript and the responds to the reviewer’s comments are as follows:

Reviewer comments:  

Abstract

Point 1: Line 66 – is: applied for identifies – should be: applied for identification?

- We changed it

Point 2: Group V ? The authors write that HCA allows for the division of the studied varieties into 4 groups. So where does group V come from?

- We changed to Ⅳ

Point 3. How was the PCA analysis optimized? Did the authors generate scree plots in PCA?

Usually, PC1 contains the largest share of variability, and the subsequent components are smaller and smaller. So what are the PC1 vs. PC3, PC1 vs PC4 etc? If they are included in the work, refer to them in the discussion.

  • Of course, we fully considered scree plot, eigenvalues(more than 1), and cumulative percentage in PCA optimize step. Generally, as reviewer mentioned, PC1,2,and 3 are main components, but our PCA results are PC1(16.9%), PC2(15.4%), and PC3(12.6%), that could be explained only 45% of total variances. And scree plot also displayed as below

Based on the scree plot results, maybe more than 10 principal component to be selected.

Therefore, we are select until PC7, which is more than 70% accumulated cumulative level, and displayed plot, that also possible to confirmed the individual PC combinations such as PC1 vs PC2, or PC1 vs PC3 like below

And as an additional opinion, when looking at the comprehensive PCA results, we think that clustering analysis is better for confirming the tendency of population characteristics.  

Point 4. Figure 5 is unreadable – so small graphs

  • We have modified Fig. 5.

Discussion (lines - 182 -190) should be expanded.

  • We expanded to PCA analysis in discussion (lines 274-279)

Round 2

Reviewer 1 Report

The authors have positively responded to my comments, I therefor rate all ratings to high for originality/novelty,  significance of content, quality of presentation, scientific soundness, interest to the readers and overall merit.

Author Response

Daer reviewer#1

On behalf of my co-authors, we thank you for giving us an opportunity to revise our manuscript, and we appreciate the editor and reviewers for your positive and constructive comments and suggestions on our manuscript entitled; Comparative Analysis of Volatile Compounds of Gamma-Irradiated Mutants of Rose (Rosa hybrida)

We have reviewed our manuscript carefully, and tried our best to improve our manuscript and made some changes in the manuscript based on the editor and reviewers’ comments. We marked the changes in red in our revised paper. We appreciate for editor and reviewers’ warm work, and hope that the correction will meet with approval. The main corrections in our manuscript and the responds to the reviewer’s comments are as follows:

Point 1: The authors have positively responded to my comments, I therefor rate all ratings to high for originality/novelty, significance of content, quality of presentation, scientific soundness, interest to the readers and overall merit.

  • Thank you

Reviewer 2 Report

in: Abstract

- it should be described what “major compounds” are.

In: Introduction

- line 57: the sentence “The content of aliphatic hydrocarbons…” is incorrect, because the acyclic terpene alcohols geraniol, nerol, and citronellol are the most important compounds in the rose essential oil.

 in: Results

- the information about the yield, color, and odor of the essential oil for each sample is missing.

- in Table 2: Data about terpenes is missing.

- in the text and in Table 2 the information about the total identified compounds is missing.

in: Discussion

- the explanation about different chemical composition of the samples is missing.

- the classification of the compounds is incorrect.

 in: Material and Methods

- the moisture content (%) of the plant, the method for its determination and the conditions of the drying should be specified. The essential oil yield should be calculated in % to absolute dry weight.

- the information about the condition of hydro distillation of the rose flowers is missing.

- it should be explained why the solvent n-hexane is used.

Author Response

Dear reviewer #2

On behalf of my co-authors, we thank you for giving us an opportunity to revise our manuscript, and we appreciate the editor and reviewers for your positive and constructive comments and suggestions on our manuscript entitled; Comparative Analysis of Volatile Compounds of Gamma-Irradiated Mutants of Rose (Rosa hybrida)

We have reviewed our manuscript carefully, and tried our best to improve our manuscript and made some changes in the manuscript based on the editor and reviewers’ comments. We marked the changes in red in our revised paper. We appreciate for editor and reviewers’ warm work, and hope that the correction will meet with approval. The main corrections in our manuscript and the responds to the reviewer’s comments are as follows:

REVIEWER COMMENTS:
Point 1: in: Abstract-it should be described what “major compounds” are.
- We changed it for abstract
Point 2: “the sentence “The content of aliphatic hydrocarbons…” is incorrect, because the acyclic terpene alcohols geraniol, nerol, and citronellol are the most important compounds in the rose essential oil.
- We have reflected the content in the introduction (line 60)
Point 3. the information about the yield, color, and odor of the essential oil for each sample is missing.
- We have reflected in result (lines 75-100)
Point 4: Data about terpenes is missing.
- Unfortunately, the top 10 volatile compounds are not including terpenes. See Table 1 for details.
Point 5: in the text and in Table 2 the information about the total identified compounds is missing.
- We added it (lines 103-106)
Point 6: the explanation about different chemical composition of the samples is missing.
- We were referred to the different chemical composition of the samples in Hierarchical cluster analysis
Point 7: the classification of the compounds is incorrect.
- We changed in the manuscript based on the Beilstein system
Point 8: the moisture content (%) of the plant, the method for its determination and the conditions of the drying should be specified. The essential oil yield should be calculated in % to absolute dry weight. the information about the condition of hydro distillation of the rose flowers is missing.
- We added it (lines 294 -298)
Point 9: it should be explained why the solvent n-hexane is used.
- We explained why the solvent n-hexane is used in discussion (lines 259-261)

Reviewer 3 Report

After changes, this manuscrit could be published in its present form.

Author Response

Dear reviewer #3

On behalf of my co-authors, we thank you for giving us an opportunity to revise our manuscript, and we appreciate the editor and reviewers for your positive and constructive comments and suggestions on our manuscript entitled; Comparative Analysis of Volatile Compounds of Gamma-Irradiated Mutants of Rose (Rosa hybrida)

We have reviewed our manuscript carefully, and tried our best to improve our manuscript and made some changes in the manuscript based on the editor and reviewers’ comments. We marked the changes in red in our revised paper. We appreciate for editor and reviewers’ warm work, and hope that the correction will meet with approval. The main corrections in our manuscript and the responds to the reviewer’s comments are as follows:

Reviewer comments:  

Reviewer #3

Point 1 After changes, this manuscript could be published in its present form.

  • We rewritten in the manuscript based on the reviewers’ comments
